

# Characterization of a complex near-surface structure using well logging and passive seismic measurements

Beatriz Benjumea[1], Albert Macau[1], Anna Gabàs[1], Sara Figueras[1]

[1]Institut Cartogràfic i Geològic de Catalunya. Parc de Montjüic. 08038 Barcelona, Spain.

*Correspondence to*: beatriz.benjumea@icgc.cat

**Abstract.** We combine geophysical well logging and passive seismic measurements to characterize the near surface geology of an area located in Hontomin, Burgos (Spain). This area has some near-surface challenges for a geophysical study. The irregular topography is characterized by limestone outcrops and unconsolidated sediments areas. Additionally, the near-

surface geology includes an upper layer of pure limestones overlying marly limestones and marls (Upper Cretaceous). These materials lie on top of Low Cretaceous siliciclastic sediments (sandstones, clays, gravels). In any case, decreasing seismic velocity with depth is expected. The geophysical datasets used in this study include sonic and gamma ray logs at two boreholes and passive seismic measurements: 224 H/V stations and 3 arrays. Well logging data defines two significant changes in the P-wave velocity log within the Upper Cretaceous layer and one more at the Upper to Lower Cretaceous

contact. This technique has also used for refining the geological interpretation. The passive seismic measurements provide a map of sediment thickness with maximum of around 40 m and shear-wave velocity profiles from the array technique. A comparison between seismic velocity coming from well logging and array measurements defines the resolution limits of the passive seismic techniques and helps for its interpretation. This study shows how these low-cost techniques can provide useful information about near-surface complexity that could be used for designing a geophysical field survey or for seismic

processing steps such as statics or imaging.

## 1 Introduction

Variation of near surface seismic properties has a significant influence on seismic reflection quality. Differences on sediment thickness over hard-rock or the presence of high velocity layers (HVL) produce strong statics and energy attenuation. This

HVL can act as a barrier for seismic energy and dramatically affects seismic signal penetration. For this reason, characterization of low velocity structures under high-velocity layers are poorly achieved by conventional seismic reflection processing. Well known cases of high-velocity structures are salt bodies (Martini and Bean, 2002) or basalt layers (Flecha et al., 2011).





Near-surface carbonate rocks are an example of high-velocity layers. Since cementation can add stiffness to shallow carbonate rocks increasing the elastic behaviour and hence velocity, velocity inversions with depth are common (Eberli et al., 2003). On the other hand, insolubles reduce the velocity in carbonates if they are more than 5% of the rock weight. Thus if the clay content of carbonates increases with depth, we find an additional origin for decrease of velocity with increase of depth. Shallow carbonates present also karstic surfaces usually filled with unconsolidated sediments adding more complexity to the near-surface. In any case, subsurface architecture mapping with seismic reflection method requires a good quality shallow velocity structure for static corrections performance. Detailed near-surface models usually are obtained from high density borehole information or velocity models derived from seismic tomography and /or first arrival inversions (Bridle et al., 2006, Ogaya et al., submitted).

The study area of this work is located in Hontomin (Burgos, Spain). This site has been the target for oil exploration during the early 70s. For that purpose, four boreholes were drilled between late 1960s and 2007 (H1, H2, H3 and H4 in Alcalde et al., 2014). Currently, new interest has been motivated in the Hontomin site as a geological storage of CO2 due to the presence of a deep saline aquifer. Different geophysical methods have been used for geological characterization of the deep structures (Ogaya et al., 2014, Alcalde et al., 2014). The near-surface geology of this site is quite complex. It includes irregular topography, surficial unconsolidated sediments in some areas combined with carbonate outcrops in others. In addition, velocity inversion in carbonate rocks is expected as previously discussed.

This paper focuses on the application of inexpensive methods (passive seismic based on seismic noise analysis) supported by well logging techniques as a good way of obtaining a detailed near-surface model. In particular, the objective of this study is to obtain a sediment map thickness of the study area, seismic characterization of carbonates and underlain sediments. This information is useful for modelling addressed to seismic survey design in such complex area and it can also help to obtain a detailed near-surface model for statics correction and for seismic imaging techniques.

## 2 Geological setting

The study area is part of the North Castilian Platform located in the southern sector of the Basque-Cantabric Domain, north of the Ebro and Duero Basins and at approximately 10 km north of the Ubierna Fault. The North Castilian Platform was formed during the Mesozoic extensional phase (Roca et al. 2011). During this period, the basin was infilled by a thick sedimentary sequence (Quesada et al., 2005). Tectonical context changed at the end of the Cretaceous age producing the basin deformation during the Pyrenees orogeny (Muñoz, 1992).



From a stratigraphic point of view, the sequence studied in this work starts with the Lower Cretaceous sediments (Purbeck facies) formed by clays, sandstones and carbonates. They have a marine-continental origin and are covered by a continental sequence including Weald facies, Escucha and Utrillas Formations. (Quintà, 2013). They are composed by alternating siliciclastic sediments (sands/gravels, low cemented sandstones and clays). The sequence ends with Upper Cretaceous

limestones as well as Lower and Upper Cretaceous marls, limestones and calcarenites outcrop in the area (Fig. 1). In addition, Miocene and Quaternary sediments including conglomerates, sands and clays cover bedrock on some sectors.

A priori, near –surface geology complexity is expected since the topography changes from Quaternary plains to carbonate outcrops. On the other hand, buried bedrock topography is suspected to be highly irregular due to the karstic processes

common in limestones. Hence, sediment thickness will be highly variable along the study area. In addition to this issue, outcrop carbonates are in general very hard in contrast to the materials covered by them. The top unit is formed by massive banks of grainstone-packstone limestones with sparitic cement and bioclastic wackestone carbonate (IGME, 1997). The next unit at the bottom includes two intervals. The first one is composed by marly limestones and marls on top of limestones and fine sediments (silts, marls, lutites). The second one is an heterogeneous interval with marls, bioclastic limestones,

calcarenites, sandy limestones, sandstones and lutites with high content of organic material. These Upper Cretaceous units lie unconformably on a Lower Cretaceous unit known as Utrillas, as described above. This unit is characterized by siliciclastic sediments with low cemented degree. With this description, a high-to-low stiffness profile (from top to bottom) is expected which adds more complexity to the near-surface structure.

**3 Geophysical methods**

**3.1 Geophysical well logging**

Geophysical well logging techniques have been applied to petroleum exploration over many decades (Serra and Serra, 2004). Borehole geophysics is an integrated part of the characterization of oil/gas reservoirs. It contributes not only to provide information about lithology and structural features but also to assess petrophysical properties of a reservoir. For much of this

period, oil logging equipment has been adapted to other objectives such as hydrogeology and mineral exploration where small-diameter boreholes are common. Nowadays, microprocessors are capable of processing signals coming from slim probes which makes unnecessary the adaptation of oil acquisition techniques to shallow applications (Paillet et al., 2004).

Several logging tools can provide different physical properties of geological strata adjacent to boreholes. In this work, we

focus on sonic and natural gamma logging. For more adequate information about logging probes the reader is referred to the literature (e.g., Ellis and Singer 2007).



### 3.2 Passive seismic (Array and H/V techniques)

The analysis of seismic noise is a valuable tool for subsurface characterization as shown by many authors such as Aki (1957) or Okada (2003). These techniques are founded on using the energetically dominant part of seismic noise which consists mainly of surface waves (Bonnefoy-Claudet et al., 2006). Two different types of methods exploit surface wave energy from
seismic noise.

First, array techniques focus on the surface wave dispersion (frequency-dependent velocity) in order to obtain a shear-wave velocity profile by inversion process. These techniques require acquiring seismic noise simultaneously on a group of seismometers. The dispersion character of surface waves can be extracted using different methods. In this work, we focus on
Frequency-Wavenumber (FK) method and Spatial Autocorrelation (SPAC) method.

### 3.2.1 Array data analysis with FK method (Horike (1985))

This method is based on the assumption that waves arrive in a plane across the receiver setup. The first step in the analysis of the signals according to the FK method is to obtain the so-called beam power plot. This is calculated after shifting the signal at each receiver according to a specific wavenumber (kx, ky), velocity and frequency. The stacking of the shifted signals is
done in the frequency domain. Repeating this process for a wavenumber, velocity and frequency values within a defined range, a complete beam power plot is retrieved. Maxima search methods of this plot provide an estimation of the travel velocity and direction of the waves.

### 3.2.2 Array data analysis with SPAC method (Aki (1957))

Another method to analyze array signals is the spatial autocorrelation method (SPAC) that assumes a random distribution of seismic sources both in space and time. Aki (1957) shown that the autocorrelation ratios between two receivers is dependent on the phase velocity and the array geometry. The application of this method uses the mean of autocorrelation ratios $\bar{\rho}$ obtained at each pair of receivers located at a distance r and considering all azimuths.

$$\bar{\rho}(r,\omega)=J_o\left(\frac{\omega r}{c(\omega)}\right) \quad \text{Eq. (1)}$$

Where $J_o$ is the zero-order Bessel function, $c(\omega)$ is the phase velocity for a certain frequency. Aki (1957) proposed circled array configuration with different radii to obtain $c(\omega)$.

Bettig et al. (2001) introduced a modification of the SPAC method that allows applying the method for different array configurations. With this modified SPAC method, the autocorrelation coefficients are obtained from station pairs separated a distance r within a certain range ($r_1$, $r_2$) or rings instead of a fixed distance. Since this method is suitable for different
configuration, the same geometry can be used to apply both FK and SPAC methods.





Dispersion or autocorrelation curves are subsequently inverted to obtain shear-wave velocity profile. Since we are dealing with 1D method, this information is assigned to the center of the array setup.

### 3.2.3 H/V technique

In addition to the array techniques, a second way of analyzing seismic noise is the horizontal-to-vertical spectral (H/V) ratio method. The H/V method computes the ratio between the Fourier amplitude spectra of the horizontal and vertical components of seismic noise measurements at a single station. This technique is based on the idea that frequency-dependent ellipticity of surface wave motion can explain the H/V spectral ratio shape. In areas characterized by sediment over hard-rock, it is widely accepted the association between the frequency corresponding to the H/V spectral ratio peak and the soil resonance frequency (Nogoshi and Igarashi (1970)). This technique was revised by Nakamura (1989) and has been proposed as a quick, reliable and low-cost technique for site-effect characterization in earthquake seismology (Lermo and Chávez-García 1993; Bard 1999). Since the 90's, several authors have introduced the H/V method as suitable for exploration studies (e.g. Ibs-von Seht and Wohlenberg 1999; Delgado et al., 2000; Benjumea et al. 2011). These studies benefit from the relationship between the soil resonance frequency ($\upsilon_0$) and bedrock depth (H) to delineate bedrock geometry on basins (Gabàs et al., 2014). A relationship between these two quantities ($\upsilon_0$ and H) includes the average shear-wave velocity of the sediments ($V_s$):

$$\upsilon_0 = \frac{V_S}{4 \cdot H} \quad \text{Eq.(2)}$$

### 4 Data acquisition and processing

#### 4.1 Geophysical well logging

During February 2012 geophysical logs were acquired in two boreholes in the study area: GW-1 up to 400 m depth and GW-3 that reached 150 m depth (Fig. 1). Two probes were used: dual induction with natural gamma sensor and three-receiver sonic probe. The logging equipment is from Robertson Geologging Ltd and includes a 500 m-winch. In this work we focus on total natural gamma values and P-wave and S-wave velocities obtained from the sonic dataset. The sonic probe has three receivers spaced 20 cm to record full-waveform data generated by a monopole source. Measurements for the two sondes were made in the upward direction.

Data processing for both records includes depth matching and 11 point median filter to remove spikes. Full-wave sonic dataset was processed to measure the formation compressional and shear-wave velocities. To obtain P-wave velocities we





use a combination of manual first-arrival identification and semblance analysis (Kimball and Marzetta, 1984). The steps of this combined processing are: filtering of the signals for DC removing, obtaining semblance plot with the three filtered signals, identification of the first maximum of semblance map corresponding to the P-wave arrival, manual identification of the first arrival at the first receiver, obtaining theoretical first-arrival time for receiver two and three using the velocity

calculated from semblance analysis, quality control of these theoretical arrivals: in case of difference between theoretical and experimental arrival adjust first arrival of first receiver or adjust maximum selection on semblance plot. This procedure helps to select a maximum on the semblance analysis in case of multiple maxima and also helps to identify first arrivals with low signal-to-noise ratio.

Regarding shear-wave velocity, traditional acoustic log measurements use semblance analysis to retrieve this value from the refracted shear wave. However, this is only possible in fast formations where shear-wave velocity is higher than P-wave velocity of the borehole fluid. If fluid velocity is higher than S-wave velocity no shear-wave arrival is detected (no critical refracted wave is generated). Some authors use Stoneley waves analysis to retrieve Vs information (e.g. Stevens and Day, 1986). In this work, the lack of low frequencies precludes the use of Stoneley waves for Vs estimation. Only some sectors of

shear-wave velocity have been obtained in both boreholes corresponding to fast formations.

## 4.2 Passive seismic survey

### 4.2.1. Array method

Previous geological information obtained from lithological description of oil well H2 (Alcalde et al., 2014) and P- and S-

wave velocity extracted from well logging in this work helps to define a seismic velocity model of the study area. This model has been input to a dispersion curve modeling to plan the array geometry.

Three 2D arrays were deployed at the test site (Fig. 1). The procedure for each array consisted in recording simultaneous seismic noise at six stations forming two equilateral triangles with two different radii to a common center where a seventh

sensor was located. One triangle was rotated 60 °respect to the other one in order to obtain good azimuthal coverage (Fig. 2). After the first recording, the inner triangle was moved to a corresponding outer triangle keeping the same centre and a second time window was acquired. This procedure was repeated increasing the distance from the triangle vertex to the centre. Array 1 and 3 used radii of 25, 55, 100, 250, 400 and 1000 m. For Array 2, the radii were 10, 25, 55, 100, 400 and 750 m. The minimum radius for this array is smaller than for Arrays 1 and 3 since a significant thickness for unconsolidated

sediments was expected. Seven Sara SL06 digitizers were connected to seven triaxial Lennartz LE-3D/5s sensors. The sensors were covered by a plastic box to reduce wind noise. The sampling rate was fixed to 200 samples per second. All stations were equipped with GPS timing.





To constitute our input dataset for retrieving shear-wave velocity, we can either use measured dispersion curves for FK method or extract autocorrelation curves using SPAC method. Both techniques have been tested and yield similar results. Data processing has been carried out using Geopsy (http://geopsy.org/).

Figure 3 shows an example of FK histograms for each triangle combination corresponding to array 1-MT 12. Dotted and continuous black lines mark the limits for resolution and aliasing obtained from maximum wavenumber and minimum wavenumber respectively. These limits are calculated from the array response for the applied geometry (Wathelet et al., 2008). Continuous black line with error bars mark the maximum of the histogram related to the dispersion of surface waves

energy of seismic noise. The combined dispersion curve corresponding to all the radii is shown in Fig. 4. Slowness slightly decreases with frequency for frequencies higher than 4 Hz approximately. This relationship indicates the presence of high velocity layers overlying lower velocity material.

Regarding SPAC processing, we have used the modification proposed by Bettig et al. (2001) calculating the spatial

autocorrelation curves on four rings of different radius ranges. An example of these curves for Array 1 is shown in Fig. 5.

An unique dispersion curve (DC) built from the DC of each configuration has been inverted using the neighbourhood algorithm that carries out an stochastic search through the model space (P-wave velocity, S-wave velocity, density and thickness layer). The algorithm was adapted and implemented by Wathalet (2008) in the DINVER software. The same

software has been used for the inversion of the autocorrelation curves obtained from the SPAC method.

The parameterised model consists of four uniform layers over half-space at the base where velocity and thickness are constrained by certain ranges. Knowing the presence of velocity inversion helps parametrization for both DC and SPAC curves inversion. We also define a coupling between P and S-wave for each layer. The obtained dispersion and

autocorrelation curves are inverted with the neighborhood algorithm generating a total of 12550 models for each technique. Only the models with a misfit less than 0.9 will be considered. The misfit function is defined according to Wathelet et al. (2008).

### 4.2.2. H/V method

H/V data acquisition has been carried out in two different field surveys: one in 2010 and the second one in 2013 with a total of 224 stations. For the first survey, a total of 150 single station measurements have been used. The stations were formed by a Spider digitize system (Worldsensing) and a 5s three-component Lennartz (LE-3D 5s). Sampling frequency was 100 Hz and record length varied from 15 to 60 minutes depending on location. For the second survey, 17 new H/V measurements in





addition to 57 datasets coming from array stations have been analysed. In this case, the station included a SARA SL06 digitizer and the same seismometer than the first survey. Sampling frequency was set up to 200 Hz and record length ranges between 20 to 120 minutes.

H/V spectral ratios were computed by dividing the noise recordings into 120 to 300 s-long windows. The resulting 224 H/V spectral ratios were analysed considering the recommendations proposed by SESAME (Bard et al., 2004).

## 5 Results and Interpretation

### 5.1 Geophysical logging and seismic velocities

Figure 6 shows the P- and S-wave logs of the GW-1 and GW3 boreholes obtained from the processing of the three receiver full-waveform records. The record corresponding to the far receiver and the gamma ray log are also included. P-wave velocity log (Vp) shows significant variation in the GW-3 borehole and up to 200 m depth in the GW1 borehole. At the shallow part, velocity log is characterized by a high-velocity layer overlying a zone with a significant velocity decrease with depth. The bottom of this high-velocity layer is 37 m depth in the GW1 borehole and 62 m depth in the GW3 borehole.

Below this zone, a high-velocity thin layer can be observed at 86 m depth at the GW1 location and at 110 m depth in the GW3 borehole. This sharp Vp change can be used as stratigraphic marker. From 92 m depth down to 200 m depth, P-wave velocity log of the GW1 borehole shows a pattern of alternating maximum-minimum velocity values. The last sector (below 200 m) at GW1 borehole is quite uniform with velocity values within a range of 3200-3600 m/s. The gamma ray response shows very low values (< 15 cps) in the shallow part up to 37 m depth of the GW1 borehole and up to 62 m depth of the

GW3 borehole. This is in agreement with the high velocity layer delineated by the sonic log. Below this layer, the gamma ray response smoothly increases down to a depth of 82 m (GW1) and 100 m (GW3). The natural gamma readings of the GW1 borehole follow an alternating pattern of maximum and minimum values from that depth down to the borehole bottom. The maximum values are located in the sector below 200 m depth.

A detailed lithological interpretation of both types of sequences is conducted based on the geophysical properties. Since boreholes drilled two different geological environments: marine (Upper Cretaceous) and continental (Lower Cretaceous), different logging datasets has been used for a detailed interpretation of each borehole.

On the one hand, Upper Cretaceous materials are mainly limestone rocks where porosity, pore types and insoluble content

control the variation of physical parameters (Eberli et al., 2003). Regarding elastic properties, there is not a direct correlation between seismic velocities and depth of burial or age. Velocity inversions with increasing depth are common such the



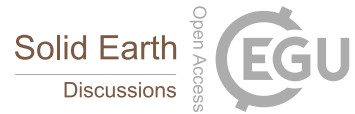

observed ones in Fig. 6. Seismic properties combined with gamma ray measurements will be used in order to characterize Upper Cretaceous rocks following the next steps:

- Zonation process based on two physical parameters (Vp and natural gamma). This process was made using WellCad software and identifies depth intervals with similar characteristics (Davis, 2002).
- Mean values calculation at each interval and for each parameter.
- Crossplots of the mean values and visual cluster recognition. Vp and natural gamma coming from GW-1 and GW-3 are merged together in a single plot (Fig. 7a). The criterion for clustering has been identifying lowest gamma values and highest P-wave velocity values. The grey ellipsoid in Fig. 7a delineates this cluster interpreted as pure or massive limestone. Fast velocities (>4000 m/s) are reached if the clay content is below 5% (Eberli et al., 2003). The
rest of the points are related to marly limestones and marls.
- Projection of these two clusters to the corresponding depth interval in the well log plot (pure limestones in blue and marly limestones and marls in purple).

On the other hand, lower Cretaceous sediments are characterized by alternating gravel/sand and clays. Hence, seismic
velocity changes are not suitable for distinguishing between layers since velocity ranges overlap for these lithological types. This fact has been confirmed by the uniformity of P-wave velocity values within this sector (Fig. 6). However, natural gamma log can help to discriminate between gravel/sands and clays (Fig. 7b). Having established three ranges of gamma values based on the histogram, the projection to the depth column of these groups has been made (Fig. 6). The ranges of gamma values have been established as: 30-55 cps (gravel/sands-yellow colour), 55-85 cps (clays/gravels mix-orange
colour) and 90-120 cps (clays-red colour).

Using the lithological description of both boreholes and based on the aforementioned geophysical parameters, we have delineated the passage from the Upper Cretaceous materials rock to the Lower Cretaceous sediments at 180 m depth at GW1 location.

### 5.2 Passive seismic

### 5.2.1 Array-shear wave velocity

Figure 8 shows the results (Vs) from dispersion (FK) and autocorrelation (SPAC) curves inversion corresponding to each array. This figure allows comparison of both techniques capabilities for shear-wave velocity characterization and contacts
delineation. Both models show a similar structure. The main characteristics are:





(i) The first layer shows low velocity at array 1 (around 500 m/s) and array 2 (150-290 m/s). For array 3, this layer is characterized by higher velocity (660-820 m/s). The thickness varies from 7-20 m at array 2 to 25 m at arrays 1 and 3.

(ii) The second layer displays high velocity (1400-2000 m/s) for the three arrays. This layer bottoms at 55-60 m (array 1), 65-70 m (array2) and 85 m (array 3).

(iii) The third layer shows the main differences between the FK and SPAC solutions for array 1. The inversion of autocorrelation curves requires the incorporation of two sub-layers to assure a good fit. The first one reaches 315 m depth and it has 870 m/s of shear-wave velocity; the second one of 1300 m/s shows a maximum bottom depth of 765 m. The FK velocity model shows a velocity of 1100 m/s and a bottom depth of 700 m for this third layer at array 1. On the other hand, shear-wave velocity is 1050 m/s (array 2) and 1100 m/s for array 2 and 3, respectively. This layer reaches 570 m (from SPAC) or 630 m (from FK) at array 2 whereas it shows a maximum bottom depth of 730 m at array 3.

(iv) The shear-wave velocity of the last layer shows different values depending on the applied array technique. It ranges from 2300 m/s to 3000 m/s. The decrease of resolution of the method makes higher the uncertainty of shear-wave velocity at this depth. However, we can expect fast formation at depths higher than 600-700 m.

Velocity inversion stands out from these models, in particular between layer 2 and layer 3.

The first layer could be interpreted as Quaternary/Tertiary sediments. The second one could be related with the Upper Cretaceous over the third one interpreted as Utrillas and probably top of Escucha sediments. Finally the last layer can be interpreted as the bottom of Escucha Formation or Weld with Purbeck facies. The shallow part of the velocity model including the velocity inversion will be compared with the logging results in the Discussion section.

**5.2.2 H/V-frequencies and sediment thickness map**

We have obtained the H/V spectral ratio for the 224 stations. The shape of these ratios can be classified in four different types.

(i) H/V spectral ratios show clear peaks that can be related to the soil resonance frequency at that station (Fig. 9a).

(ii) Multiple peaks can be identified. This H/V morphology is associated to multiples impedance contrasts and/or 2D/3D bedrock structure. In both cases, it is difficult to assign a soil resonance frequency although adjacent stations or other geophysical results can help to define that frequency (Fig. 9b).

(iii) Flat H/V spectral ratio indicates that the station is located on bedrock (Figure 9c).

(iv) H/V spectral ratio is characterized by a constant amplification over a wide frequency range, mainly at the high-frequency range (> 1 Hz), forming a step pattern. This constant amplification is not related to subsoil structure but probably to anthropic seismic noise and not to soil response (Figure 9d). For this reason, we do not use the results corresponding to this type of H/V shape.



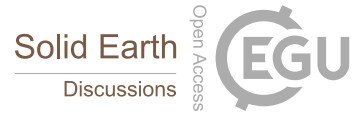

H/V frequency at each station corresponds to the frequency at the H/V spectral ratio maximum. Figure 10 shows the frequency values color coded on a map view from group (i). It also includes the location of the sites identified as rock sites (flat H/V spectral ratios-group ii) and the sites corresponding to an H/V step pattern.

We distinguish three zones with clear H/V peaks: two sectors in the NE and SE quadrants with fundamental frequencies ranging between 2 and 8 Hz and a third sector in the South part with frequencies values between 4 and 10 Hz. In all the sectors, the fundamental frequency gradually decreases with distance to the zone center. These sectors are related to soft soils with a significant thickness and impedance contrast with bedrock. Shallow bedrock or rock outcrops are located in the

NW quadrant, SE end and central part of the study area.

In this study, GW3 location has been used as ground-truthing for H/V peak interpretation, i.e., to find which lithological can cause this fundamental frequency in the study area. One H/V station was located next to GW3 with the result of a fundamental frequency of 4.3 Hz. According to the GW3 lithological interpretation, a first layer of 20 m thickness composed

by quaternary and Miocene sediments (clays and silts) overlies Upper Cretaceous limestone. This contact causes a significant impedance contrast that produces the observed fundamental frequency of the soil. On the other hand, we can use this value to estimate a shear-wave velocity for the unconsolidated sediments that helps to produce sediment thickness maps. Using equation (1), a shear-wave velocity value of 344 m/s is obtained. As presented in the array results, the Vs range of unconsolidated sediments is 290 m/s – 500 m/s. The value obtained at GW3 applying H/V technique is included within this

range. Hence, Vs varies along the study area as expected from the presence of Quaternary and Tertiary sediments of different lithology. We have used the limits of the Vs range from arrays measurements to estimate a range for sediment thickness (Fig. 11). For the production of this map, the natural neighbour gridding method has been used. This method is suitable for datasets with high density in some areas and paucity in others. Maximum thickness for 500 m/s shear-wave velocity varies between 40 and 45 m (Figure 11a) and for 290 m/s varies from 20 to 25 m (Fig. 11b).

**6 Discussion**

One of the limits of vintage well logging data is the dead zone in the sonic logs corresponding to the near surface. In the study area, three oil exploration boreholes lack the first 400 m of sonic data (Alcalde et al., 2014). In this framework, slim hole well logging is key to get information of the shallow part. Both Vp and Vs has been calculated for the shallow Upper

Cretaceous limestone (first 90 m depth) and a complete P-wave velocity log is retrieved from the sonic log for the first 400 m. The velocity information increases understanding of physical properties of Upper Cretaceous materials and Utrillas (Lower Cretaceous). It will also help to interpret velocity information obtained with the array technique. On the other hand,



the combination with gamma ray log has enabled us to refine the geological interpretation of this shallow part. This interpretation constrained the results for the other geophysical results obtained in this study.

Regarding the passive seismic methods, the H/V technique can provide a detailed sediment thickness map also required for resolving near-surface issues. When borehole information is sparse, the H/V technique can be a good alternative for obtaining a shallow subsurface model for statics calculation or field survey planning.

The array technique results helped to identify a significant velocity inversion in the first 100 m depth. The bottom of the high-velocity layer has been identified between 60 and 80 m approximately depending on the array location and method of surface waves analysis (FK or SPAC). If we compare these results with the shear-wave velocity log obtained at GW1 and GW3 (Fig. 12), we observe that the bottom of the high-velocity layer would correspond to the change from a sector characterized by velocity decrease with depth to another one made by interbedded layers of high and low velocity. The influence of this type of layering in a seismic signal with a wavelength longer than the fine-scale details of velocity variations has been studied by numerous authors (e.g . Stovas and Ursin, 2007). According to Hovem (1995), for a large ratio between wavelength ($\lambda$) and thickness ($d$) of one cycle in the layering, the layered sector behaves as a homogeneous, effective medium (Backus 1962). For low ratio, the alternate layers may be replaced by thicker single layer using a time-average or ray velocity for the total depth range.

In our case, seismic wavelength of the surface waves analyzed with the array technique varies from 125 m at 50 m to 500 m at 200 m according to the dispersion curve (Fig. 4). On the other hand, spatial Fourier analysis of the GW1 sonic log from 92 to 200 m gives two maximum at 10 and 18 m. In our case, the $\lambda/d$ ratio would range between 5 at 92 m to 25 at 200 m. Since these values would correspond to different models (Backus average or time-average velocity), we have considered both models.

Firstly, a P-wave time-average velocity has been obtained from 92 to 200 m giving a value of 2348 m/s. This is in agreement with a velocity inversion between the near-surface limestone and the layers below (marly limestones and marls). Since S-wave velocity is not available for this sector we cannot obtain an S-wave time average velocity.

Secondly, from a Backus average point of view, we need first to define the number of layers and assign a P- and S-wave velocity characteristic of the high and low velocity layers. For a characteristic thickness of 10 m, we can use a number of 10 layers with alternate high and low velocity as equivalent to the interbedded layer sector (92 to 200 m). P-wave for the high velocity layers has been fixed to 4000 m/s and for the low-velocity layers to 1500 m/s. This leads to a Backus P-wave velocity of 1986 m/s. Regarding S-wave velocity, we have chosen a high-velocity of 2000 m/s confirmed by Vs calculation from sonic log where some hints of shear-wave velocity were obtained at the high-velocity layers (e.g. at around 150 m

depth of GW1). For the low-velocity layers, we have used the Poisson's ratio value (0.31) obtained by Dvorkin et al. (2001) from rock physics analysis of well log data in marls. This value corresponds to Vp/Vs ratio of 1.91 hence leading to a shear-wave velocity of 790 m/s. Using the Backus average equation for shear-wave velocity, we obtain a velocity of 1040 m/s. This equivalent layer velocity is similar to the one obtained from the array technique and would explained the observed

velocity inversion.

However, the passage from Upper Cretaceous to Lower Cretaceous materials has not been resolved by the array technique. The time-average or the Backus average for the P-wave velocity is of the same order than the one obtained for the Low Cretaceous rocks (average velocity of 2300 m/s). Hence the effective medium corresponding to the interbedded sector is

indistinguishable of the Utrillas sediments from a seismic array point of view. P-wave velocity log enables the detection of this change associated with the Upper to Lower Cretaceous transition.

When we compare the S-wave velocity values obtained from well logging and the array technique, we observe that array velocity is always lower than S-wave logging velocity. This result is consistent with the fact that sonic logging uses high-

frequency signals which travels at different velocity than low-frequency seismic signals (Box and Lowrey, 2003). Due to the dispersion effects, the pulses generated with a sonic probe travel a few percent faster than the array surface waves.

## 7 Conclusions

Slim well logging completes the depth record of old well logging data with sonic data from the surface down to 400 m depth. This information combined with the gamma ray log helps to refine geological interpretation of the first 400 m. P-wave velocity data delineates two significant changes with depth in the Upper Cretaceous materials: a high velocity layer overlying a zone with velocity decrease with depth, and an interbedded sector with high and low velocities. For the propagation of a seismic wave, this sector is equivalent to a low velocity layer. Below this layer, we find low-velocity

sediments corresponding to Utrillas (Low Cretaceous) formation.

H/V technique enabled us to obtain a sediment thickness map of a complex near-surface area in a fast and affordable way. This methodology can produce good results in areas with scarcity of boreholes.

Shear-wave velocity profile up to 800 m is obtained with the array technique. Regarding the near-surface, this profile maps a velocity inversion within the Upper Cretaceous that has been correlated to the presence of the high-velocity layer over the interbedded sector. The change from Upper to Lower Cretaceous sediments has not been delineated with this technique.



As a first step, passive seismic methods combined with near-surface geophysical logging constitute a fast way of characterizing near-surface complexity. Identification of velocity inversions and sediment thickness map are valuable inputs for geophysics field planning, seismic statics correction or further modelling.




**Appendix A.**

Natural gamma loggings from GW1, GW2 and GW3 with the comments made by Dr. Andrés Pérez-Estaún.





**Author contribution**

S. Figueras and A. Gabàs planned the field surveys for well logging and acquired the data from GW3 borehole. They have also participated with comments and ideas for processing and interpretation. A. Macau planned the field surveys, acquired

well logging data from GW1 borehole, the passive seismic datasets and processed the H/V and array data. He carried out the inversion of the dispersion and autocorrelation curves and interpretation of the shear-wave velocity models. B. Benjumea planned the field surveys, acquired well logging data from GW1 borehole and passive seismic datasets. She analyzed the well logging data and processed the sonic datasets. She has interpreted the datasets and made the comparison between well-logging and passive seismic results. She prepared the manuscript with contributions from all co-authors.

**Acknowledgments**

We would like to thank Andrés Pérez-Estaún in memoriam for being the person who invited us to participate in this project. He contributed with enthusiasm and wisdom to the interpretation of well logging data (Appendix A).

We acknowledge all the people who have assisted in the field: I. Marzán, F. Bellmunt, E. Falgàs, J. Rovirò, M. Esquerda, J. Salvat and M.A. Romero. This work has been partly funded by the Spanish Science and Innovation Ministry project PIER-

CO2 (Progress In Electromagnetic Research for CO2 geological reservoirs CGL2009-07604).



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





**FIGURES**

5    Figure 1 Surficial geological map of the study area (Quintà. 2013) with locations of the H/V measurements, the array centers and the boreholes discussed in this paper. Overview map of the Iberian Peninsula with location of the study area.





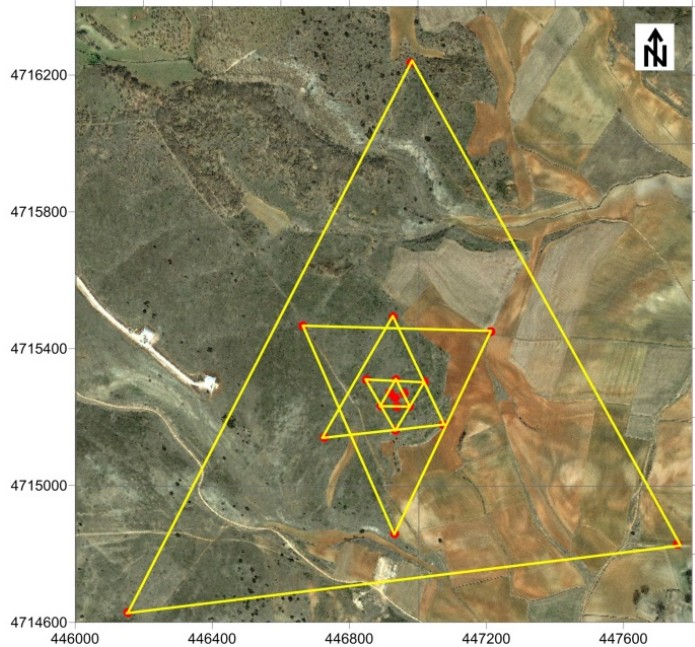

Figure 2. Geometry used for the seismic noise measurements at array 1. Red circles mark the position of the stations and yellow lines show the triangle setup. Map is shown in UTM coordinates, datum ETRS89.




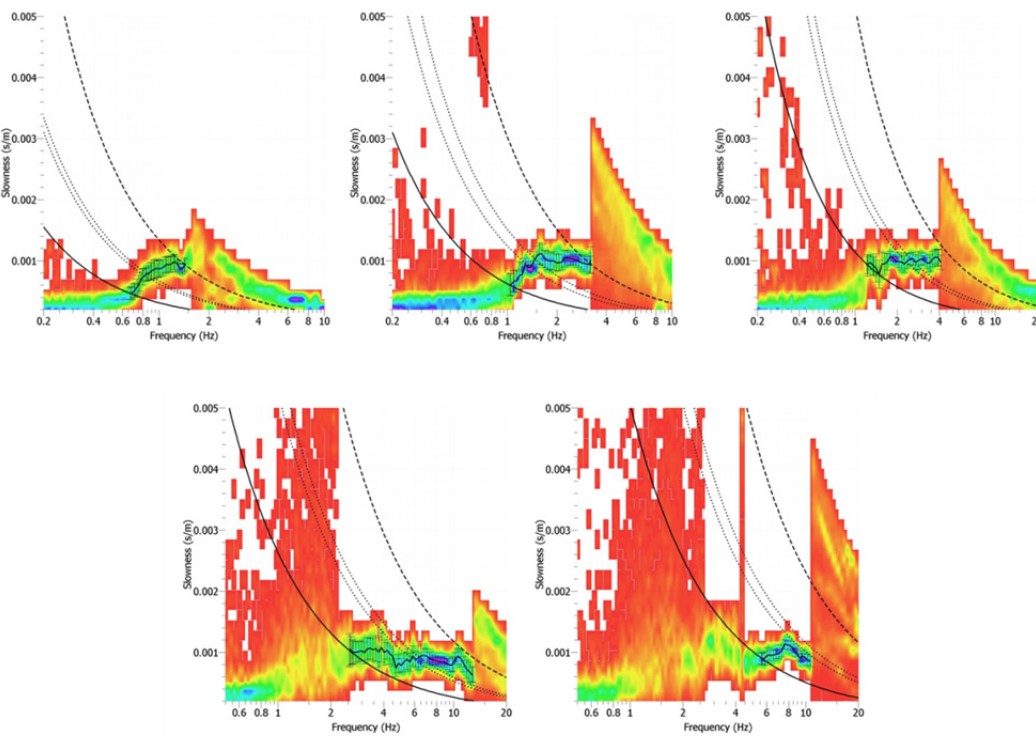

Figure 3. Histograms of velocities obtained from frequency-wavenumber analysis for each triangle configuration performed for Array 1. For each plot the black line with error bars mark the dispersion curve. The continuous line indicates a constant wavenumber value of kmin/2 related to the resolution limit of this array geometry (Wathelet et al., 2008). The dashed line is a constant wavenumber value of kmax that defines the aliasing limit constrained by the array geometry. The dots represent kmin (left) and kmax/2 (right) respectively.





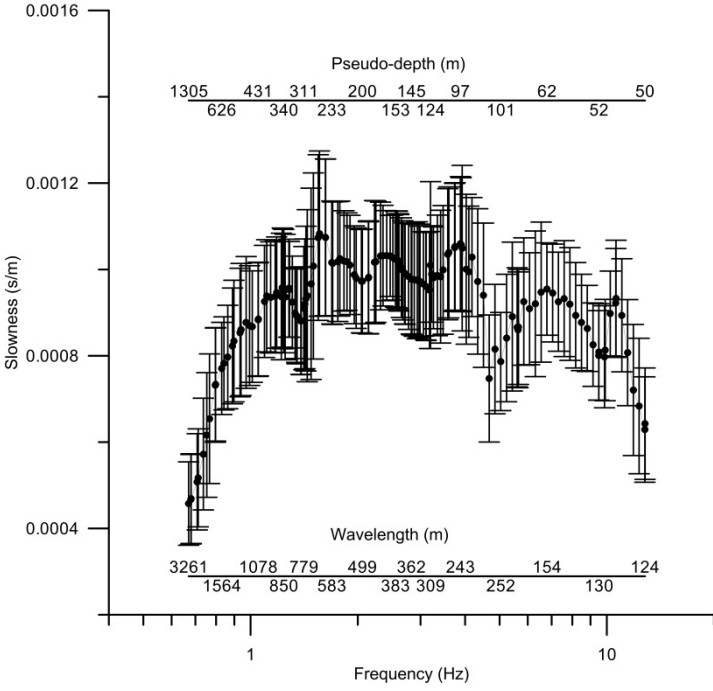

Figure 4. Total dispersion curve for Array 1 obtained from the combination of the dispersion curves shown in Figure 3. The corresponding wavelength for each slowness and frequency values is indicated in a bottom auxiliary axis as well as the pseudo-depth (top auxiliary axis) calculated as wavelength/2.5.

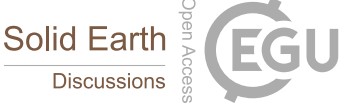



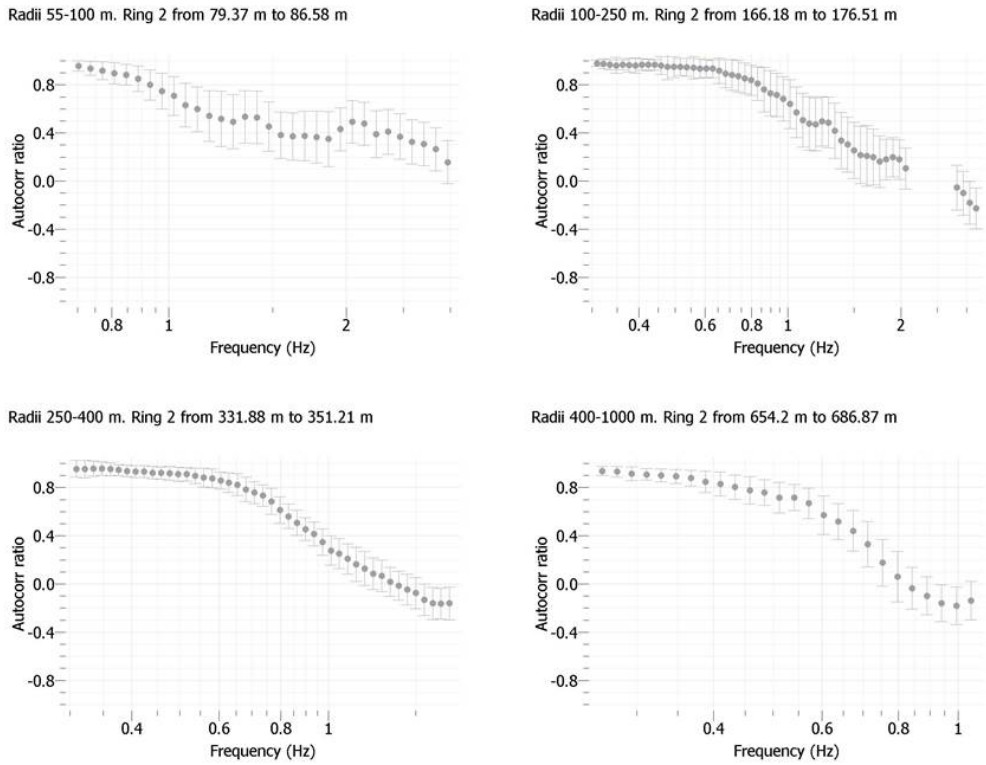

Figure 5. SPAC analysis. Autocorrelation curves (grey dots with error bars) obtained for Array 1 at each ring.



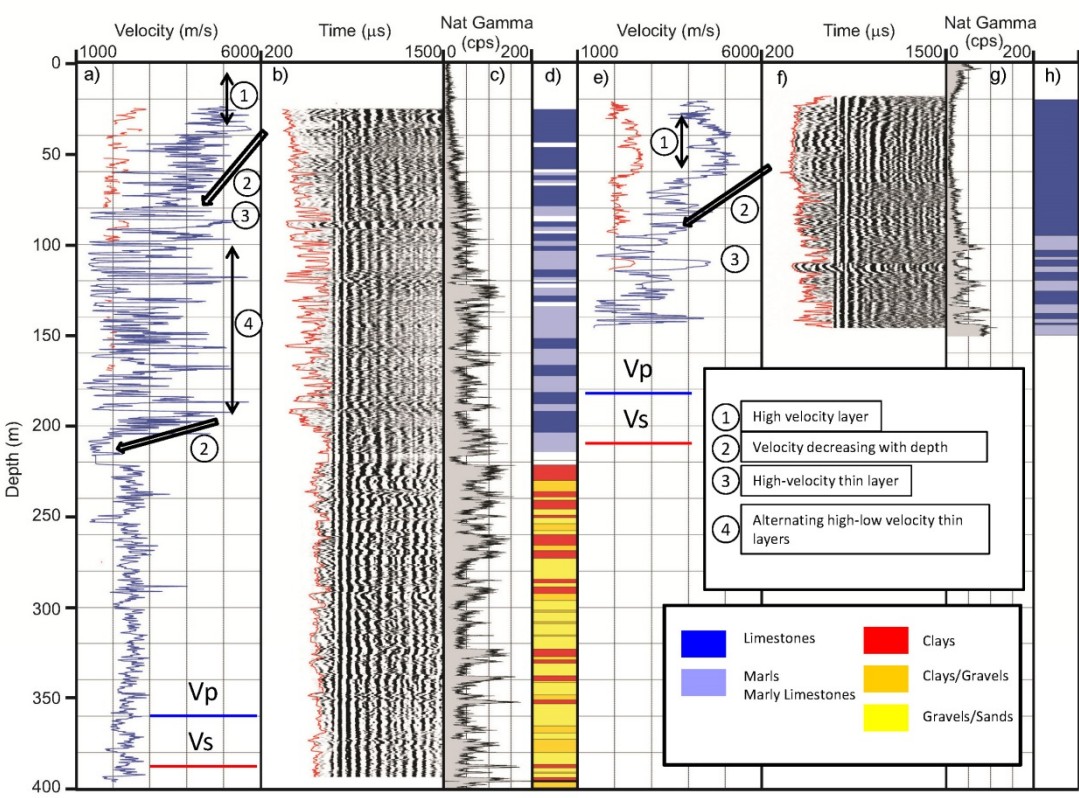

Figure 6. a) P-wave velocity (blue line) and S-wave velocity (red line) for GW-1, b) Recorded sonic waveform for the far-receiver from GW-1 borehole c) natural gamma log from GW-1 and lithological interpretation resulting from zonation process and clustering as shown in Figure 7 for GW-1. e), f), g) and h) same as a), b), c) and d) respectively but for GW-3.



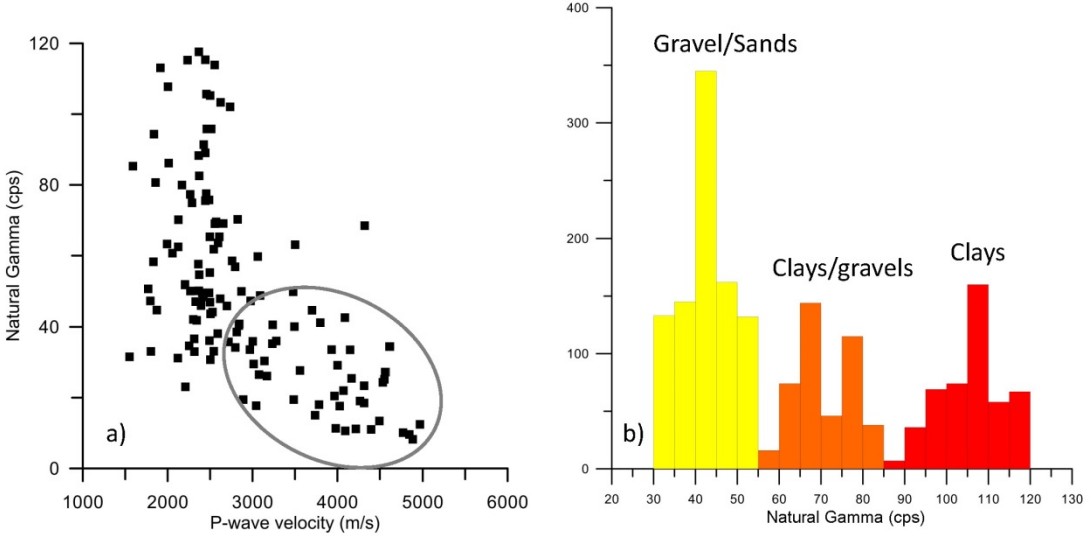

Figure 7. a) Cross-plot of the average P-wave velocity and natural gamma values for each zone defined with the zonation process. The plot includes measurements from both GW-1 and GW-3 corresponding to the Upper Cretaceous sector. The grey ellipsoid encircles the points characterized by high P-wave and low natural gamma values interpreted as limestone. The points outside this area has been related with marly limestones and marls b) Distribution of natural gamma values for Lower Cretaceous materials from GW-1 with the geological interpretation of the three histogram groups.





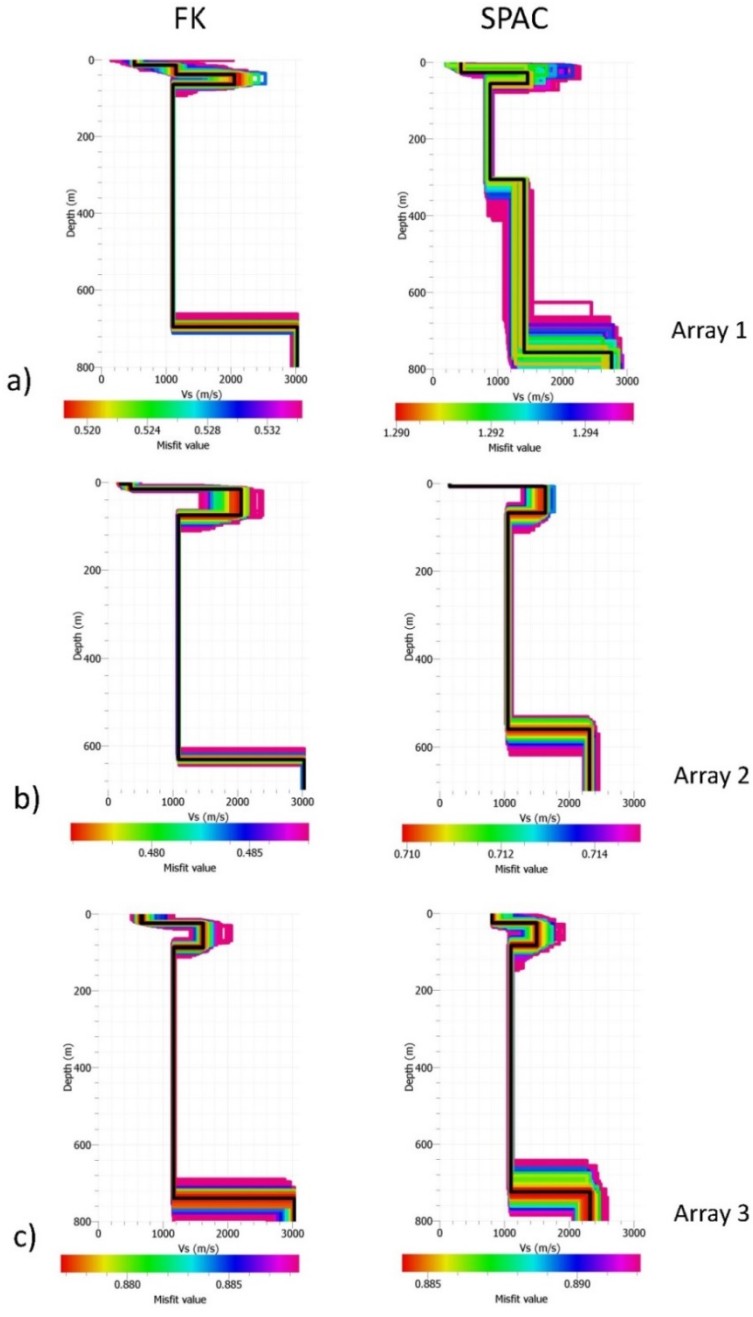

Figure 8. Shear-wave velocity models obtained from the dispersion curve inversion (FK method- left) and the autocorrelation curves inversion (SPAC method – right) array technique for: a) Array 1 b) Array 2 c) Array 3. The models have a misfit lower than 0.9. Colour code shows the misfit of each model. The black line represents the shear-wave velocity model with minimum misfit.



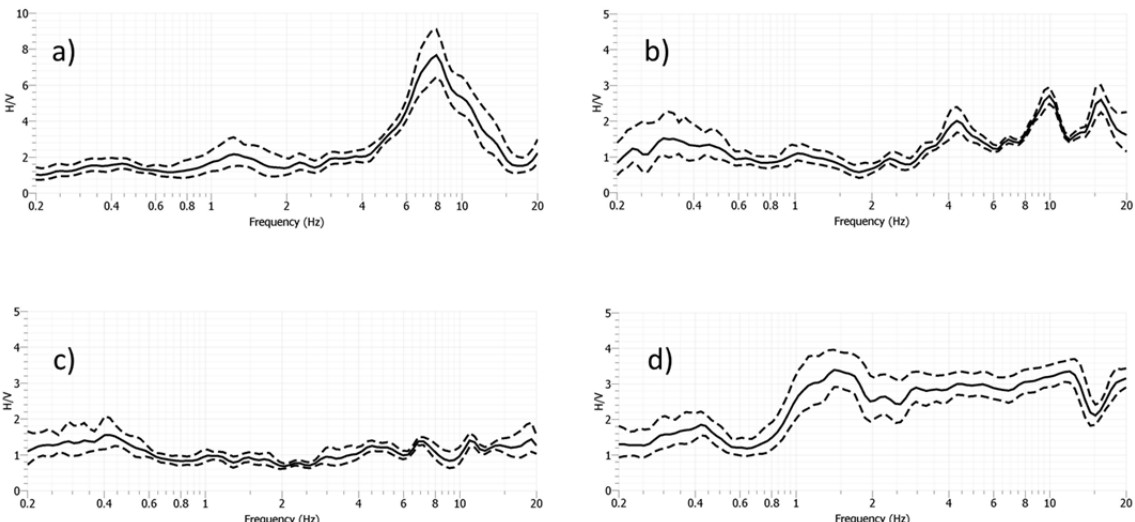

Figure 9. H/V topologies observed in the study area. a) station with a clear peak b) station with multiple significant peaks c) station without peak d) H/V ratio showing constant amplification over a wide frequency range (step shape).





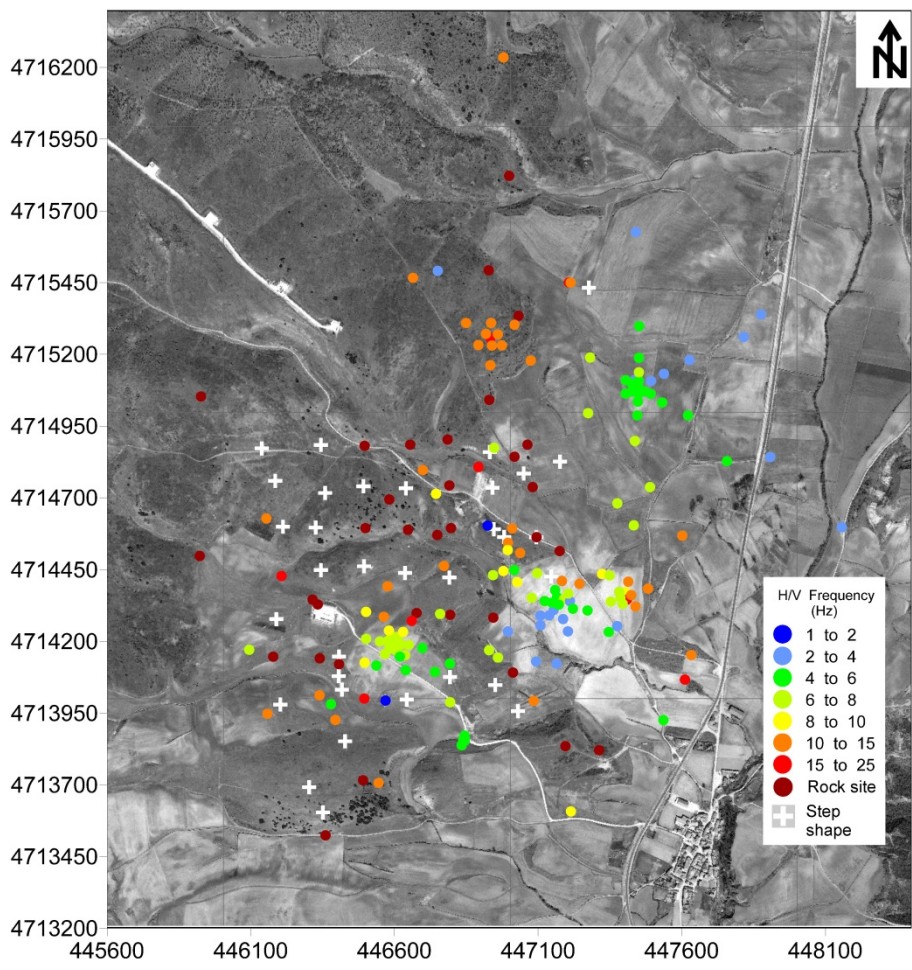

Figure 10. H/V method. Fundamental H/V peak frequencies over the study area (colour coded). White crosses denote stations with step morphology (Figure 9).





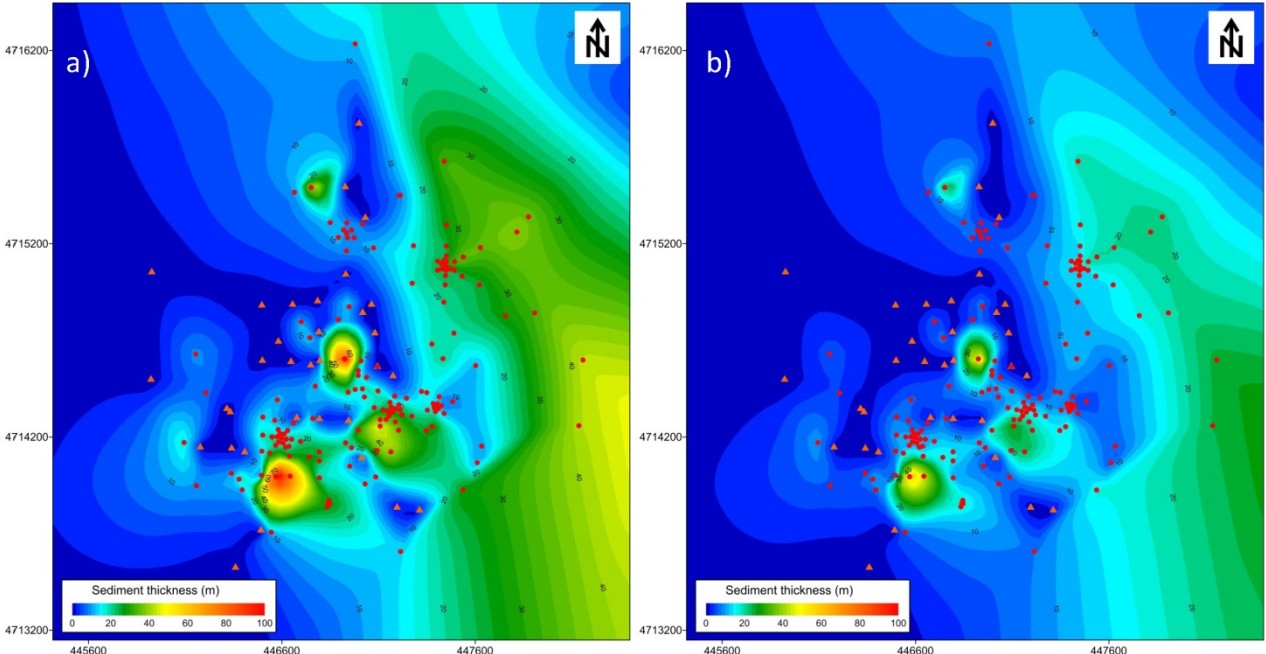

Figure 11. Contour map showing the sediment thickness obtained from interpolation of thickness values obtained from H/V frequencies (Figure 10) using a) Vs=500 m/s and b) 290 m/s.





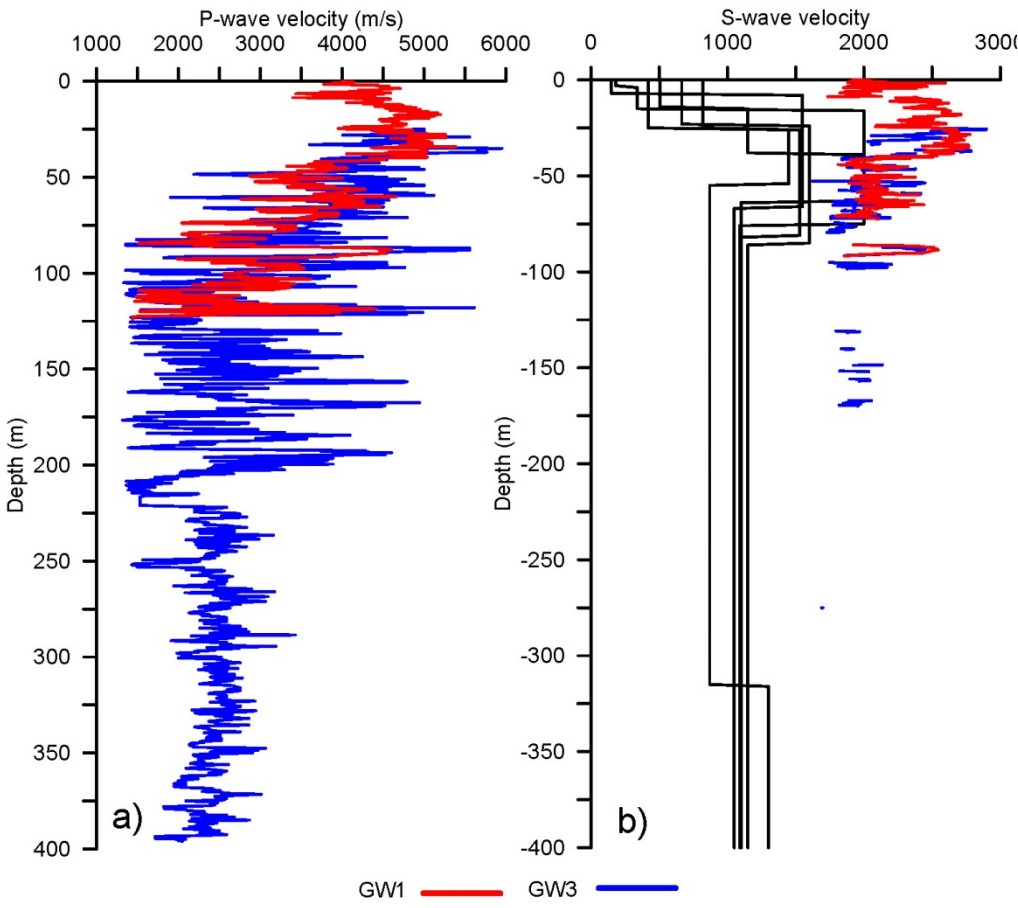

Figure 12. a) P-wave velocity logging for GW1 borehole (blue line) and GW3 (red line). b) S-wave velocity logging for GW1 borehole (blue line) and GW3 (red line). Shear-wave velocity obtained from the array technique (black lines).