# Peer review of "Characterization of a complex near-surface structure using well logging and passive seismic measurements"

_Solid Earth, 2016_

## Referee Comment (RC1) · Anonymous Referee #1 · 9 Mar 2016

Manuscript "se-2016-19" by Benjumea et al. presents a shallow site characterization of a location in Hontomin, Burgos, Spain, using geophysical methods. Among the methods used are well logging (gamma log and sonic log for P- and S-wave velocities) as well as passive seismic array methods using ambient noise (shear velocities via dispersion curves (FK-method), autocorrelation (SPAC-method), and H/V method). The compact study is an application of well-established methods and thus an (engineering) showcase of using different geophysical method to derive subsurface structure (also to combine logging information and surface-base passive seismic methods). The paper does not deal with method development

The study area is dominated by a complex structure and carbonate rocks. Oil explo-

ration took place in the past (availability of boreholes), however, current interest in the region is related to CO2 sequestration. One interesting complication of the subsurface is a surficial high velocity layer and a velocity inversion at greater depth.

The paper is clearly structured, the written English is (with some smaller exceptions) good (see below). The paper presents new data. The individual methods seem to be properly and sufficiently explained. The majority of the figures is good, however, some figures need to be improved (see below). The abstract is informative and concise.

Overall, I think that the study is interesting from the method-point-of-view (combination/ comparison of different techniques), however, it is a rather specific study focusing on a particular location (limited regional/local scope). I suggest to add some more critical notions regarding the suitability of the particular passive seismic methods which would be interesting for researchers planing similar studies elsewhere. In conclusion, I think that the manuscript is worth to be published in Solid Earth after some moderate revision.

Major issues:

1) P2, L19-24, P12, L6, and P14 last sentence: The authors state that results from the passive methods could be used for statics corrections (I assume for P-wave reflection seismics). Even if the depths are correct (however, they have a rather large uncertainty; see below) the P-velocities are also needed (however, not provided by these methods). So, its use for statics corrections might be rather small. Furthermore, I doubt that the depth resolution e.g. of the H/V method is accurate enough for statics. For example, you need a good value of the shear velocity to convert frequencies to depths (which might be difficult to obtain). In any case, also the other methods (FK & SPAC) have rather uncertainties for layer depths (easily visible e.g. in Figure 8). Please add some more discussion and/or mention these potential limitations.

2) Could you say something about the conversion of the group velocity (derived from the dispersion) to layer velocity (material property)? Maybe also important for the last

paragraph of the Discussion section discussing differences between the velocity derived by sonic log and those by array techniques.

3) P3, L24ff: Why is the information regarding "adaptation of oil acquisition techniques to shallow applications" important? If it is important for the article, please provide more information. If not, please skip this information. In any case, it's difficult to understand in the current form. Are "oil acquisition" and "oil logging" the correct terms?

4) Figures 3 and 8 seem to be direct printouts of the program code and are of poor quality. Labels/annotations are too small; color scale in Figure 3 missing. Please revise.

5) Figure 9: Labels too small. Please revise.

6) Figure 11: I strongly suggest to only show interpolated values in regions with data coverage! Please fade out (or leave out - white) regions without data points. Furthermore, some of the high values in these plots seem to be caused by just one data point – please check (or remove if outlier).

7) Figures 1, 2, 10, and 11: Please add a scale. Additionally, please state in a label or in the caption what kind of coordinates are shown (I assume UTM?).

8) Section 3.2.1: From this text I cannot understand how you derive the plots in Figure 3. Please add information.

9) Could you add some conclusions which of the passive seismic methods are – based on this study – better suited and which not. Summarizing strengths and weaknesses of the individual methods? Could be interesting for readers planning similar studies (last paragraph of conclusion section).

Minor issues:

P1, L11: "...decreasing seismic velocity with depth is expected.": Well, velocities increase again for some larger depths... Its just a layer with reduced velocity. Please

rephrase.

P2, L3: please add "fluids" after "insolubles" (the latter is just an adjective)

P2, L29: Please add "The" before "Tectonical".

P3, L11: "outcropping" instead of "outcrop"

P4, L21: Please insert "has" before "shown".

P5, L8 ff.: I suggest to rephrase the sentence "In areas characterized...": "In areas characterized by sediment over hard-rock, the H/V spectral ratio peak frequency is associated to the soil resonance frequency (Nogoshi and Igarashi, 1970).

P7, L3: please insert "the" before "SPAC"

P11, L13: Please insert "logs of the" before "GW3" and replace "location" by "borehole".

P11, L13: Please replace "with the result of" by "which showed"

P11, L24: Please replace last sentence by: "Maximum thickness of the layer with 500 m/s shear-wave velocity varies between 40 and 45 m (Figure 11a) and of the layer with 290 m/s between 20 and 25 m (Fig. 11b)

P11, L31: Please insert "our" before "understanding"

P4, L15: Remove "a" before "wavenumber", and insert "specific" instead.

P13, L7: Replace "passage" by "transition"

---

## Author Comment (AC1) · 14 Mar 2016

We thank the comments and suggestions to improve the manuscript made by Referee 1. In this file, we will reply (italic format) the major issues included in the revision. Minor changes will be added to the last version of the manuscript as well as improvement of the figures quality.

Major issues:

1) P2, L19-24, P12, L6, and P14 last sentence: The authors state that results from the passive methods could be used for statics corrections (I assume for P-wave reflection seismics). Even if the depths are correct (however, they have a rather large uncertainty; see below) the P-velocities are also needed (however, not provided by these methods). So, its use for statics corrections might be rather small. Furthermore, I doubt that the depth resolution e.g. of the H/V method is accurate enough for statics. For example, you need a good value of the shear velocity to convert frequencies to depths (which might be difficult to obtain). In any case, also the other methods (FK & SPAC) have rather uncertainties for layer depths (easily visible e.g. in Figure 8). Please add some more discussion and/or mention these potential limitations.

*Reply: Yes you are right. H/V method can only partially help to statics calculation with bedrock depth values. However, we think that in an area with a great change in this value (such as a karstic area) and with scarcity of well information, H/V become a useful tool as a first approach to assess a range of depth variation. In addition it can help to locate areas where a high-resolution refraction survey would be critical for statics calculation. We will adjust the sentences regarding statics correction to make clear that the contribution of H/V is as a first approach and only for bedrock depth.*

*The Vs estimation can be done using array techniques or use H/V method in a place with a known bedrock depth (e.g. close to a well with a lithological log). In page 11, we explain the use of both approaches for obtaining a range of Vs for sediments of the area. A H/V station close to a borehole with known geology has been used as quality control of the shear-wave array measurements. Decreasing uncertainty of the Vs for the first layer and bedrock depth would require array measurements with lower minimum distances between sensors or an active surface wave survey. Anyhow, we think that this study shows the potential of these techniques to obtain bedrock depth in a fast and effective way.*

2) Could you say something about the conversion of the group velocity (derived from the dispersion) to layer velocity (material property)? Maybe also important for the last paragraph of the Discussion section discussing differences between the velocity derived by sonic log and those by array techniques.

*The surface wave velocity derived by array techniques is the phase velocity. In order to obtain shear-wave velocity, an inversion of the dispersion curve (phase velocity versus frequency) is performed. The basis was developed for suggesting that S-wave velocities fundamentally control changes in Rayleigh-wave phase velocities for a layered earth model (Xia et al. 1999). In the software used (Geopsy www.geopsy.org), the forward computation of the dispersion curve is done using Dunkin's formulation (1965) to link parameters (Vp, Vs, density and thickness) of a stack of layers with the dispersion curve. The inversion is done using a Monte Carlo approach (neighborhood algorithm, Sambridge 1999).*

3) P3, L24ff: Why is the information regarding "adaptation of oil acquisition techniques to shallow applications" important? If it is important for the article, please provide more information. If not, please skip this information. In any case, it's difficult to understand in the current form. Are "oil acquisition" and "oil logging" the correct terms?

*The development of small probes for near-surface applications became important for the spreading of well logging surveys for other applications other than oil exploration. It allowed to acquire geophysical well logging datasets to academia or government centers without using the sophisticated and expensive*

*oil equipment. Maybe it is out of the scope of the article, hence we will skip that in the last version of the manuscript.*

4) Figures 3 and 8 seem to be direct printouts of the program code and are of poor quality. Labels/annotations are too small; color scale in Figure 3 missing. Please revise.

*Thank you for the comment. We will improve the quality of these two figures for the final version of the manuscript.*

5) Figure 9: Labels too small. Please revise.

*Same as 4. Label size will be increased for the final version of the manuscript.*

6) Figure 11: I strongly suggest to only show interpolated values in regions with data coverage! Please fade out (or leave out - white) regions without data points. Furthermore, some of the high values in these plots seem to be caused by just one data point – please check (or remove if outlier).

*Yes, we completely agree with that. The zone out of H/V stations must be white. Regarding the one-data point zones: In these sectors the good data quality assures us to keep them as useful information in spite of the paucity of measurements.*

7) Figures 1, 2, 10, and 11: Please add a scale. Additionally, please state in a label or in the caption what kind of coordinates are shown (I assume UTM?).

*Scale will be added to these figures. The coordinates are UTM ETRS89.*

8) Section 3.2.1: From this text I cannot understand how you derive the plots in Figure 3. Please add information.

*Section 3.2.1 will be completed. According to Wathalet (2005), "the horizontal velocity in the FK method is calculated for different frequency bands. The raw signals are first divided in short time windows the length of which may depend upon the considered frequency. A Fourier transform is calculated for the signal of each sensor after a proper cutting of the current time window. The FK transformation is calculated in the frequency domain on the cut signals. FK analysis assumes horizontal plane waves to travel across the array sensors. Considering a direction and velocity of propagation, the relative arrival times are calculated at all sensor locations and the phases are shifted according to these times. The semblance is calculated by the summation of shifted signals in the frequency domain." The last step is to locate the maximum of the semblance that gives us an estimation of the velocity and the azimuth of the plane wave across the array. For each time window and for each frequency we have a maximum corresponding to certain propagation velocity. Figure 3 is a compilation of histograms of the pair frequency and velocity obtained from the semblance maximum search at each time window. Each histogram corresponds to a subarray setup (one sensor in the center and six sensors located at two different radii from that center). Obviously a color scale of the histograms is missed. We will add that for the last version of the article.*

9) Could you add some conclusions which of the passive seismic methods are – based on this study – better suited and which not. Summarizing strengths and weaknesses of the individual methods? Could be interesting for readers planning similar studies (last paragraph of conclusion section).

*In general, passive seismic methods are suitable for areas with seismic noise or with logistical issues (no space for instrumentation setup, problems with source regulations). In addition, these techniques are cost-effective. H/V is fast, cheap, suitable as a reconnaissance step. (no need of great space for instrumentation setup less than 1 square-m) It can help to detect areas with strong variation in bedrock depth and obtain an estimation of this value with additional information. The use of the*

*array techniques to obtain shear-wave velocity information allows to increase the investigation depth of active surface waves technique. Also they are suitable in areas with inversion velocity such as this one, overcoming the limitation of refraction surveys for instance. FK and SPAC are based on different assumptions in the surface waves propagation. Their comparison is useful as quality control. Comparison between these two approaches can be found in different works (Wathelet et al., 2008, Cadet et al., 2011)*

*References*

*Cadet, H., Macau, A., Benjumea, B., Bellmunt, F., & Figueras, S. (2011). From ambient noise recordings to site effect assessment: the case study of Barcelona microzonation. Soil Dynamics and Earthquake Engineering, 31(3), 271-281.*

*Dunkin, J. W. (1965). Computation of modal solutions in layered, elastic media at high frequencies. Bulletin of the Seismological Society of America, 55(2), 335-358.*

*Sambridge, M. (1999). Geophysical inversion with a neighborhood algorithm—II. Appraising the ensemble. Geophys. J. Int, 138(3), 727-746.*

*Wathelet, M., Jongmans, D., Ohrnberger, M., & Bonnefoy-Claudet, S. (2008). Array performances for ambient vibrations on a shallow structure and consequences over V s inversion. Journal of Seismology, 12(1), 1-19.*

*Xia, J., Miller, R. D., & Park, C. B. (1999). Estimation of near-surface shear-wave velocity by inversion of Rayleigh waves. Geophysics, 64(3), 691-700.*

---

## Referee Comment (RC2) · Anonymous Referee #2 · 28 Mar 2016

General remarks on the paper titled "Characterization of a complex near-surface structure using well logging and passive seismic measurements" by Beatriz Benjumea, Albert Macau, Anna Gabàs, Sara Figueras-SE-2016-19.

Authors proposed a method that is used geophysical well logging and passive seismic measurements to characterize the near surface geology of the area located in Hontomin, Burgos (Spain).

They used sonic and gamma ray logs at two boreholes and 224 H/V stations and 3 arrays of passive seismic measurements:

The authors obtained and declared that passive seismic measurements provide a map

of sediment thickness with maximum of around 40 m and shear-wave velocity profiles from the array technique.

Passive seismic methods are very useful technique, especially if there are deep sediments and need to be higher depth resolutions. In normal case, active seismic measurements and techniques such as MASW is enough to obtain 30-40 meter penetration depths for geotechnical studies. The resolution and penetration depths also depend on the source which is used in the measurements in the field. In this study, authors applied the passive seismic techniques and also used well log data. But there are some important points that they have to be care to obtain more accurate results. It is also very important what kind of initial data and technique is used in the study. These parameters affect the final results directly.

In the study, when we analysis the figure 12, it is clearly seen in the first 50m and 100 meters, there are very big differences in the P and especially S wave velocity distributions. It means that obtained results are more accurate and trustable in deeper parts but not successfully obtained in near surface approximation. There are 300-500 m/sn velocity gap among the near surface calculations. Therefore authors must discuss why they can not obtain accurate results in the near surface depths?

I advise them to re-analysis the used methods or check the initial parameters used in the study. The obtained results also affect the obtained sediment thickness maps?

I should do advise them to read the following studies which are very close to the current study and it can increase the quality of the paper and enrich the study for the readers.

Kanli A.I., Kang T., Pinar A., Tildy P., Pronay Z., 2008, A Systematic Geophysical Approach For Site Response of The Dinar Region, Southwestern Turkey", JOURNAL OF EARTHQUAKE NGINEERING, vol.12, pp.165-174.

Singh S., Kanli A.I., 2016, Estimating shear wave velocities in oil fields: a neural network approach, GEOSCIENCES JOURNAL, vol.20, pp.221-228.

Kanli A.I., 2009, Initial Velocity Model Construction of Seismic Tomography In Near-Surface Applications, JOURNAL OF APPLIED GEOPHYSICS, vol.67, pp.52-62.

Kanli A.I., 2011, Integrated Approach for Surface Wave Analysis from Near-Surface to Bedrock, in: Advances in Near-Surface Seismology and Ground-Penetrating Radar, R.D. Miller, J.D. Bradford and K. Holliger, Eds., Society of Exploration Geophysics, Tulsa, pp.461-475.

Kanli A.I., Tildy P., Pronay Z., Pinar A., Hermann L., , 2006, V-S(30) Mapping And Soil Classification For Seismic Site Effect Evaluation In Dinar Region, Sw Turkey, GEO-PHYSICAL JOURNAL INTERNATIONAL, vol.165, pp.223-235.
* * *